# ConAnomaly: Content-Based Anomaly Detection for System Logs

**DOI:** 10.3390/s21186125

**Published:** 2021-09-13

**Authors:** Dan Lv, Nurbol Luktarhan, Yiyong Chen

**Affiliations:** College of Information Science and Engineering, Xinjiang University, Urumqi 830046, China; lvdan@stu.xju.edu.cn (D.L.); chenyiyong578@163.com (Y.C.)

**Keywords:** log anomaly detection, log sequence encoder, LSTM

## Abstract

Enterprise systems typically produce a large number of logs to record runtime states and important events. Log anomaly detection is efficient for business management and system maintenance. Most existing log-based anomaly detection methods use log parser to get log event indexes or event templates and then utilize machine learning methods to detect anomalies. However, these methods cannot handle unknown log types and do not take advantage of the log semantic information. In this article, we propose ConAnomaly, a log-based anomaly detection model composed of a log sequence encoder (log2vec) and multi-layer Long Short Term Memory Network (LSTM). We designed log2vec based on the Word2vec model, which first vectorized the words in the log content, then deleted the invalid words through part of speech tagging, and finally obtained the sequence vector by the weighted average method. In this way, ConAnomaly not only captures semantic information in the log but also leverages log sequential relationships. We evaluate our proposed approach on two log datasets. Our experimental results show that ConAnomaly has good stability and can deal with unseen log types to a certain extent, and it provides better performance than most log-based anomaly detection methods.

## 1. Introduction

With the increase of many people’s needs, the complexity of modern systems is increasing day by day. The more complex the system, the greater the likelihood of vulnerabilities that an invader may exploit to launch attacks. As a result, anomaly detection has become an important task in building trusted computer systems [1]. An accurate and effective anomaly detection model can reduce abnormal damage to the system, which is very important for business management and system maintenance. Logs are widely used to record important events and system status in operating systems or other software systems. Since system logs contain noteworthy events and runtime states, they are one of the most valuable data sources for anomaly detection and system monitoring [2].

Logs are semi-structured text data.One of the important tasks is anomaly detection in logs [3]. It is different from computer vision [4,5,6], digital time series [7,8,9] and graphic data [10]. In fact, the traditional way of handling log anomalies is very inefficient.Operators manually check system logs based on their domain knowledge by matching regular expressions or searching keywords (such as error and Failure). However, this anomaly detection method is not suitable for large-scale systems.

More and more works start to apply schemes to process the logs automatically. Existing log-based system anomaly detection methods can be roughly classified into two categories: one is based on log event indexes, such as PCA [11], Invariant Mining [12], Deeplog [13], and QLLog [14]. The other is based on log templates, such as LogAnomaly [15] and LogRobust [16]. Although both of these two methods first parse the logs, there are two differences: one is that the log event index-based method converts the log to the event index, while the log template-based method removes the numeric information in the log to obtain the log invariant (event template). For instance, the log template of log “Received block blk_7503483334202473044 of size 233,217 from /10.250.19.102” is “Received block * of size * from *”. The other is that the first method is to encode the log event index number (e.g., using a one-hot encoding), and the other is to vectorize the log template. Although the event template-based methods can utilize semantic information in log messages compared to the event index-based methods, they cannot handle the log template that has not been seen. Moreover, both methods are highly dependent on the log parser [17,18,19,20,21], especially for log event index-based approaches. The performance of the log event index-based approach degrades significantly when the log parsers are incorrect [22].

Although log templates are structured, they are still text data. Most of the input of machine learning models needs to be digital data, not text. Therefore, extracting the features of the log template or deriving its digital representation is the core step. Meng et al. [23] form the log event vector by the frequency and weights of words. The log event vector is transformed into the log sequence vector as the input of the anomaly detection model. The transformation from word vector to log event vector or log sequence vector is called coordinate transformation. However, the frequency and weight of words ignore the relevance between words. Recently, More and more works start to apply natural language processing (NLP) methods for the log event vectorization, especially word2vec, which generates word vectors based on the positional relationship of words. However, if word2vec is directly used in the system log template, it will generate a huge word space and cause unnecessary waste of resources. Therefore, this article has made corresponding improvements to the model.

In this article, we propose ConAnomaly, an anomaly detection method that takes advantage of both the semantic relationships of log messages like the template-based method and the sequential relationships between logs. In the ConAnomaly model, we improve the word2vec [24,25,26] model to obtain the log2vec that preprocess logs. It vectorizes the log content and removes invalid information through part of speech tag [27]. Finally, multi-layer LSTM [28] and other models are used for anomaly detection. We evaluated our proposed method on BGL [11] and HDFS [29] datasets. Experimental results show that ConAnomaly is versatile and has excellent detection performance.

The key contributions of this article can be summarized as follows:We use the part of speech of the vocabulary as the standard for preliminary filtering of the log content, which reduces unnecessary waste of computing resources. To the best of our knowledge, our work is the first to utilize this to weight features.This study provides new insights to handle unseen log templates and reduce the dependence on the log parser on the market.We proposed ConAnomaly, which considers the semantic information in the log message into the log sequential anomaly detection, which improves the detection performance to a certain extent.

The rest of this article is organized as follows. We introduce the related work in Section 2 and present the theory of our work in Section 3. Besides, an overview of our scheme has two main components: log2vec and a model for anomaly detection. Finally, we evaluate the performance of the proposed model in Section 4 and conclude this work in Section 5.

## 2. Related Work

Log-base anomaly detection mainly consists of three steps: log parsing, feature extraction, and anomaly detection. We review the related works for each step.

### 2.1. Log Parsing

Log parsing extracts the log template or log event from the raw log. Figure 1 shows the parsing result of a raw log “081109 203518 143 INFO dfs.DataNode$DataXceiver: Receiving block blk_-1608999687919862906 src: /10.250.19.102:54106 dest: /10.250.19.102:50010” that come from the HDFS dataset.It is parsed into log template “Receiving block <*> src: /<*> dest: /<*>” and event “E5”. Here ‘<*>’ is a wildcard to match parameters.

There have been many studies on log parsing, e.g., Drain [21] and Spell [30]. Drain is an online log parsing method based on a fixed depth tree. When a new raw log message arrives, Drain preprocesses it using simple regular expressions based on domain knowledge. Then search for a log group, that is, for the leaf node of the tree by following specially designed rules encoded in the number of internal nodes. If the appropriate log group is found, the log messages will match the log events stored in that log group. Otherwise, a new log group is created based on the log message. It achieves high performance compared to many other log parser methods. The spell is an LCS-based [31] online stream processing log parsing method for structured stream parsing of event logs. It can dynamically accept log input, process the input in real-time, and constantly generate new log templates. In addition, He et al. Designed and implemented a parallel log parser (POP) on Spark, a large data processing platform.The original logs were divided into constants and variables, and the same log events were combined into the same cluster group through hierarchical clustering.

### 2.2. Feature Extraction

Extracting the feature of logs is the basis of anomaly detection. Generally, researchers select features from system logs, including log templates, event occurrences, event index, log variables and encode them through one-hot encoding or other weight methods. Lin et al. [2] parsed logs into log events using the log abstraction technique and convert them to vectors. Log sequences were represented as a vector of weight in an N-dimensional space after calculating the weight for each event, where N is the number of unique events. In DeepLog [13], besides the log events, it also considers the variant characteristics in the logs. Hua et al. [32] modeled the sample data as Hermitian positive-definite (HPD) matrices, and the geometric median of a set of HPD matrices is interpreted as an estimate of the clutter covariance matrix (CCM). Then By manifold filter, a set of HPD matrices are mapped to another set of HPD matrices by weighting them, that consequently improves the discriminative power by reducing the intra-class distances while increasing the inter-class distances.

In addition, more and more work start applying natural language processing(NLP) methods to log preprocessing, such as bag-of-words [33], TF-IDF [34], and word2vec.

He et al. [35] form the event count vector for each log sequence by counting the occurrence number of each log event, whose basic idea origins from bag-of-words. Lin et al. [2] propose an approach named LogCluster which turns each log sequence into a vector by Inverse Document Frequency (IDF) and Contrast-based Event Weighting. Meng et al. [15] propose a framework to model a log stream as a natural language sequence. They propose a novel, simple feature extraction method, template2vec, to extract the semantic information hidden in log templates by a distributional lexical-contrast embedding model (dLCE). The word vector is transformed to the log event vector. In this way, the semantic relationship of logs can be learned effectively.

### 2.3. Anomaly Detection

The existing anomaly detection methods based on log data are mainly classified into three categories, which are graph model-based [36,37,38], probability analysis-based [39] and machine learning based detection methods [40]. Anomaly detection based on graph is used to model the sequence relationship, association relationship, and log text content. The anomaly detection based on probability statistics adopts correlation analysis, comparison, etc., to calculate the correlation probability between log and anomaly.

At present, the machine learning-based method mainly utilizes the LSTM model to infer whether the log is abnormal or not by judging the log sequential relationships. Deeplog [13] leverages LSTM to model the sequence of log keys for a particular type of log, automatically learning normal patterns from normal log data to identify system exceptions. References [41,42] analyze the application of various LSTM models in anomaly detection, such as bidirectional LSTM and stacked LSTM.

### 2.4. Limitation of Previous Models

The limitations are as follows:The existing log-based anomaly detection system is very effective, which mostly depends on the existing log parser tools. If the tool is not available for the current log data set, the model may not perform well. Moreover, they cannot handle unknown log events or templates. In DeepLog, it utilizes Spell, an unsupervised streaming parser that parses incoming log entries in an online fashion based on the idea of the longest common subsequence (LCS), to preprocess log files. Its input for classification is a window w of the h most recent log keys. That is, w = m${}_{t-h}$, …, m${}_{t-2}$, m${}_{t-1}$, where each m${}_{i}$ is the log key from the log entry e${}_{i}$. However, if an undefined log instance is printed in a real-time environment, there is a risk that the model will crash or make incorrect predictions.Logs as unstructured data have two characteristics: one is that there is a temporal relationship between logs, which is a manifestation of the workflow; The second is that the log itself has semantics. But most of the tools available take advantage of only the first feature of logs in the anomaly detection part. For example, in LogCluster, the clustering method is leveraged to cluster log sequences that are similar in sequences.

In this paper, we propose ConAnomaly, which utilizes both the semantic and sequential relationships of logs to detect anomalies. Our approach also addresses the limitations of the previous approaches to some extent. For example, in most previous detection models, if the incoming log is slightly different from the defined log template, it will be treated as unknown data. However, since our method is to build a database of the vocabulary of the log content, most of the vocabulary used in the log content is not very different, the occurrence of unknown data will be greatly reduced.

## 3. Design of ConAnomaly

### 3.1. Overview

The overview of ConAnomaly is shown in Figure 2. The first step is log parsing, which extracts the log content from the original log files and then removes the numbers and punctuation marks in the log by regular matching. Each log is a set of words in the semi-structured text. Inspired by word2Vec, we propose a digital method, log2vec, which effectively converts the obtained log invariant into a vector sequence. The detail of this model will be described in the next section. The last step is to utilize the multi-layer LSTM to learn the log sequential relationship and to conduct anomaly detection through modeling.

### 3.2. Log2vec

Word2vec, also known as word embeddings, turns words in natural language into dense vectors that computers can understand, and maps words that have similar meanings to nearby locations in the vector space [43]. However, it cannot be directly used in the vectorization process of the logs. Firstly, some words in the log are invalid words, such as “to” and “can”, but it cannot filter them. Second, it targets at vocabulary and cannot vectorize sentences. Last but not least, if the log is not filtered, the word vector space will be large, which will be an unnecessary waste of computing resources. Therefore, we make some improvements to word2vec. We propose log2vec, a sentence representation method, which can get rid of the invalid word in logs and effectively construct log vectors.

As shown in Figure 3, log2vec includes three steps in the learning stage: (1) use the word2vec model to vectorize the word in the log content. (2) Tag the words in the obtained lexicon with part of speech. The vectors of words with part of speech labels ‘CC’, ‘TO’, ‘IN’ and ‘MD’ (as shown in Table 1) were set as zero vectors. (3) Calculate log vectors by weighted averaging the vectors of the words in the corresponding log invariants.

Moreover, the log vectors in the BGL dataset are serialized by batch processing. However, this approach cannot be leveraged directly to the HDFS dataset, because the number of logs in the HDFS log block is different. Therefore, before batch processing, it is necessary to truncate or padding the log blocks in HDFS.

### 3.3. Log Anomaly Detection

The flow of anomaly detection is shown in the solid line box in Figure 2. We firstly leverage multilayer LSTM to learn sequential relationships between logs, then use the full connection(FC) [44] layer to make the linear transformation of the learning results and map them to the label space, and finally use the softmax layer to do normalization processing.

#### 3.3.1. Lstm

The long short-term memory model (LSTM) is a popular recursive neural network structure, which has been proved to be able to predict data sequence effectively. As shown in Figure 4, LSTM controls cell state by three gates, which are respectively called the forgotten gate, input gate, and output gate.

A forgotten gate is a sigmoid unit that leveraged to determine what information needs to be discarded in the cell state. It outputs a vector between 0 and 1 by operating on *h*${}_{t-1}$ and *x*${}_{t}$. The value of 0 to 1 in this vector indicates what information in the cell state is retained or discarded. 0 means no reservation, 1 means all reservations. The formula is as follows:(1)$$f_{t}=\sigma \left(W_{f}*\left[h_{t-1},x_{t}\right]+b_{f}\right)$$

The input gate and the candidate cell choose what new information to add to the cell state. First, an input gate is used to determine which information to update, and then *h*${}_{t-1}$ and *x*${}_{t}$ are used to obtain new candidate cell information through a tanh layer that may be updated into the cell information. The formula is as follows:(2)$$i_{t}=\sigma \left(W_{i}*\left[h_{t-1},x_{t}\right]+b_{i}\right)$$
(3)$$\tilde{C_{t}}=tanh\left(W_{c}*\left[h_{t-1},x_{t}\right]+b_{c}\right)$$

A part of the old cell information is forgotten by the forgotten gate selection while a part of the candidate cell information is added by the input gate to get the new cell information *C*${}_{t}$. Equation (Equation 4) is the update operation.
(4)$$C_{t}=f_{t}*C_{t-1}+i_{t}*\tilde{C_{t}}$$

After updating the cell state, the final output of the model is obtained through the operation of the output gate. The formula is as follows:(5)$$o_{t}=\sigma \left(W_{o}*\left[h_{t-1},x_{t}\right]+b_{o}\right)$$
(6)$$h_{t}=o_{t}*tanh\left(C_{t}\right)$$

#### 3.3.2. FC

The core operation of a fully connected network is a matrix-vector product.
(7)y=W∗X+b

The essence of this layer is the linear transformation from one feature space to another. This layer is used to make a weighted sum of the features of the previous design, turning the hidden layer space back into the label space.

#### 3.3.3. Softmax

The softmax function [45], also known as the normalized exponential function, aims to show the results of multiple classifications in the form of probability.

Assumed that there has an array Y, and *y*${}_{i}$ represents the ith element in Y, then the softmax value of this element is:(8)$$S_{i}=\frac{y_{i}}{\sum_{j}y_{j}}$$

## 4. Experiment

In this section, we first describe the experimental dataset and evaluation metrics, and then compare the performance of ConAnomaly on large system log data with existed methods. Lastly, we investigate the performance impact of various parameters in this model.

### 4.1. Experiment Setting

#### 4.1.1. Datasets

We conduct our experiments on two datasets: the HDFS dataset and the BGL dataset. The summary statistics for the two datasets are listed in Table 2. The followings are the detailed information for the log datasets:BGLThere are 4,747,963 logs in the BGL dataset, which are collected from a BlueGene/L supercomputer system at Lawrence Livermore National Labs. Each BGL log was manually labeled as either normal or anomalous, and 348,469 logs were anomalous.HDFSThe HDFS dataset consists of 11,175,629 logs collected from more than 200 Amazon EC2 nodes that run Hadoop-based jobs. Program execution in the HDFS system usually involves a block of logs. Based on this theory, 575,061 blocks of logs are obtained, among which 16,838 blocks were labeled as anomalous by experts. Unlike BGL data, HDFS logs have identifiers recorded for each job execution.

**Table 2 sensors-21-06125-t002:** Summary of log data.

Datasets	#Time Span	#of Logs	#of Anomalies
HDFS	38.7 h	11,175,629	16,838 (blocks)
BGL	7 months	4,747,963	348,469 (logs)

In the following experiments, for both datasets, we first separate the normal and anomalous logs, and then, 80% of logs are extracted from both types of logs as the training data (according to the timestamps of logs), the rest are the testing data. Moreover, to solve the problem of data imbalance, SMOTE algorithm is used to synthesize the data.

#### 4.1.2. Baselines

We compare ConAnomaly with five unsupervised baseline methods. These methods are briefly described as follows:LogCluster: This article proposes an approach that clusters the logs to ease log-based problem identification. Besides, it utilizes a knowledge base to check if the log sequences occurred before.DeepLog: It proposes DeepLog, a deep neural network model utilizing Long Short-Term Memory (LSTM), to model a system log as a natural language sequence.LogAnomaly: LogAnomaly is a framework to model a log stream as a natural language sequence. It can detect both sequential and quantitive log anomalies simultaneously, which has not been done by any previous work.LogRobust: LogRobust extracts semantic information of log events and represents them as semantic vectors. It utilizes an attention-based Bi-LSTM model to detect anomalous log sequences.HitAnomaly: This work proposes a log-based anomaly detection model utilizing a hierarchical transformer structure to model both log template sequences and parameter values.

#### 4.1.3. Implementation

All the experiments are conducted on a Windows machine with Intel Core 3.40 GHz CPU and 8 GB memory. ConAnomaly is implemented through Pytorch [46] and we refer to the results from the corresponding literature for the three baseline methods.

#### 4.1.4. Evaluation Metrics

Precision, recall, and F1-score are used to evaluate the accuracy of anomaly detection methods. Precision shows the percentage of true anomalies among all anomalies detected; Recall measures the percentage of anomalies in the dataset being detected; F1-score is the harmonic average of the two indexes.
(9)$$Precision=\frac{TP}{TP+FP}$$
(10)$$Recall=\frac{TP}{TP+TN}$$
(11)$$F1\text{-}score=\frac{2*Precision*Recall}{Precision+Recall}$$

TP (True Positive) refers to the real case, that is, the real situation is positive, and the predicted situation is also positive.

FP (False Positive) means false positive example, the real situation is negative, and the predicted situation is positive.

FN (False Negative) refers to a false negative example, that is the real situation is positive, and the predicted situation is negative.

### 4.2. Evaluation on BGL Dataset

Figure 5 shows the performance of ConAnomaly compared to five baseline methods over the BGL dataset. ConAnomaly achieves the highest recall among the six methods, having an F1 score of 0.98. LogAnomaly and ConAnomaly can detect anomalies with a more than 95% F1-score, which demonstrates that the semantic information of the log is helpful for log anomaly detection. LogAnomaly generates more false alarms than ConAnomaly because it does not address the problem that the new log does not match the old log template and that the log data is not balanced. LogCluster does not achieve good detection accuracy on BGL data. The poor performance of LogCluster is caused by the high dimensional sparsity of the event count matrix. As a result, log clustering makes it difficult to distinguish between anomalies and normal conditions, which often results in a large number of false positives.

At the same time, we found that the BGL dataset has certain particularity. Table 3 shows a partial digitized BGL log sequence. The numbers in the table represent the different types of log events, for example, ’149’ represents logs that can be extracted as “External Input Interrupt (.*) (.*) (.*) tree Receiver (.*) in Resynch mode” (as shown in Table 4). As you can see from Table 3, the logs in the BGL dataset are highly stacked, which means that logs of the same type are always repeated consecutively. Based on this phenomenon, we believe that the BGL dataset is not very representative. Furthermore, we wanted to explore the detection capabilities of ConAnomaly on datasets with identifiers, so we did a similar experiment on the HDFS dataset.

### 4.3. Evaluation on HDFS Dataset

#### 4.3.1. Experiment Result

Figure 6 shows the performance of ConAnomaly over the HDFS dataset. ConAnomaly achieves the best accuracy among those methods and HitAnomaly has an F1-score of 0.98 as the second. Both ConAnomaly and HitAnomaly utilize word vectors. However, HitAnomaly transforms the log template into a fixed dimensional vector, while ConAnomaly vectorizes the contents of the logs.

In fact, many existing detection methods perform well on HDFS datasets (over 90%). This is mainly due to the log parser, which extracts the log template on the HDFS dataset very accurately. Because of this, most log processing methods use a log parser, such as the rest of the five methods in Figure 6 except ConAnomaly.

#### 4.3.2. Analysis of ConAnomaly

We first investigate the effect of window size and the number of layers on performance in ConAnomaly. As shown in the following figures, we varied the value of one parameter while using the default values for the others (the control variates method), and reported the results over the HDFS dataset.

Figure 7 shows the influence of the number of LSTM network layers (n-layer) in ConAnomaly. We observe that when the value of n-layer is greater than 2, the ConAnomaly model is not very sensitive to a different number of layer settings and the detection performances of the model are almost the same when n-layer = 2 and n-layer = 6. However, a greater number of parameters lead to a longer training time and prediction time. With this factor in mind, we choose a smaller n-layer (n-layer = 2).

Figure 8 presents the performance impact of window size in ConAnomaly. Window size refers to the maximum distance between the current and predicted word within a log. As can be seen from the figure, in general, the precision of ConAnomaly is fairly stable with concerning different window sizes while it has an impact on the recall of the model. When the window size value is equal to 3, the model has the highest recall value. The smaller the window size, the worse the semantic relevance of the logs the system learns. The larger the size of the Widow, the easier it will be to overfit the model.

#### 4.3.3. Experiment Based on the Unseen Logs

In this section, we evaluate the robustness of our model based on unseen log types. The log types are compared by the final representation of the different block_id log sequences.

First, we explore the log distribution rules of the HDFS data set. As can be seen from Figure 9 and Table 5, the number of log types increases rapidly between the first 10% and 40% of data ordered by the log timestamp. After random shuffling, the log types are close to linear distribution. That means log data of HDFS have an obvious update before 50% data. The HDFS data is suitable for evaluating the robustness of the model.

As shown in Table 6, data of 1%, 10%, 20%, and 50% in the dataset were respectively adopted as the test set, and then the number of log types in the training set, test set, and those not in the training set but appearing in the test set (the unseen log types) were counted. The results showed that the F1-score increased as the unseen log types decreased. At the same time, we observe that when the training data accounts for 1%, the detection performance of ConAnomaly is also higher than 90%, which indicates that our model has good stability and can deal with unseen log types to a certain extent.

However, in the experiment, we find that ConAnomaly presents the phenomenon of a single category of prediction when it predicts data, such as the predicted results are all normal. When the training data accounts for 10%, this phenomenon occurs 22 times. For this limitation of the model, we will conduct further research.

## 5. Conclusions

This article proposes ConAnomaly, an anomaly detection method that takes advantage of both the semantic relationships of log messages like the template-based method and the sequential relationships between logs. We designed a novel log sequence encoder to obtain log sequence representations and built its classification model based on the lstm mechanism. We evaluated our proposed method on two log datasets. Our experimental results demonstrate that ConAnomaly has outperformed other existing log-based anomaly detection methods and has a strong versatility.

One of our future work directions is to incorporate the structure of the attention mechanism into the task of log-based anomaly prediction, and we may consider the parameters in logs.

## Figures and Tables

**Figure 1 sensors-21-06125-f001:**
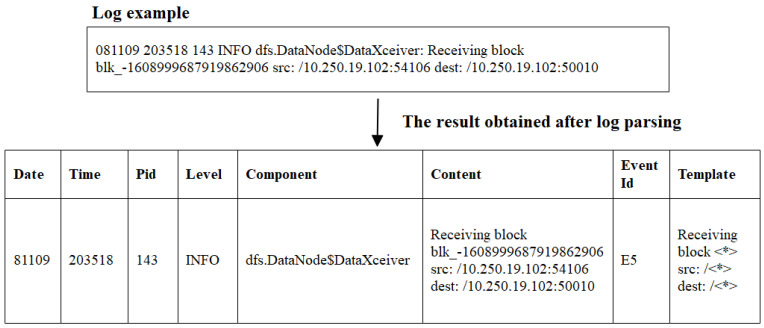
Example of log parsing.

**Figure 2 sensors-21-06125-f002:**
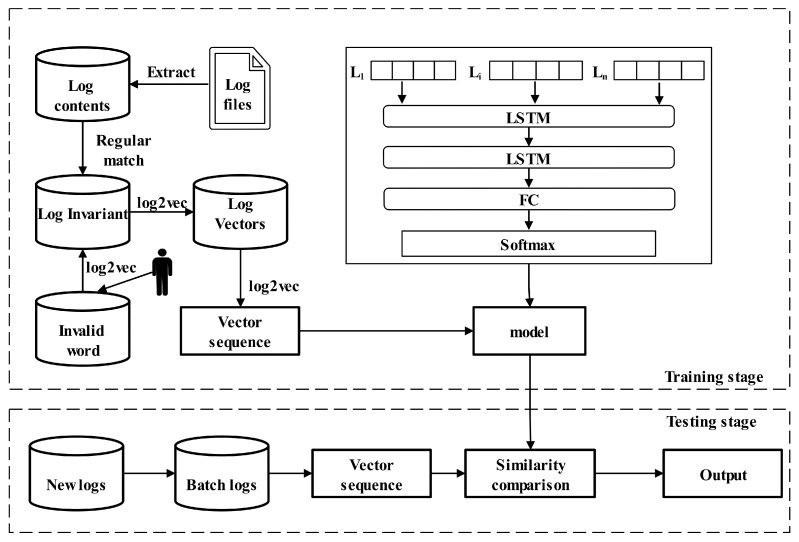
The framework of ConAnomaly.

**Figure 3 sensors-21-06125-f003:**
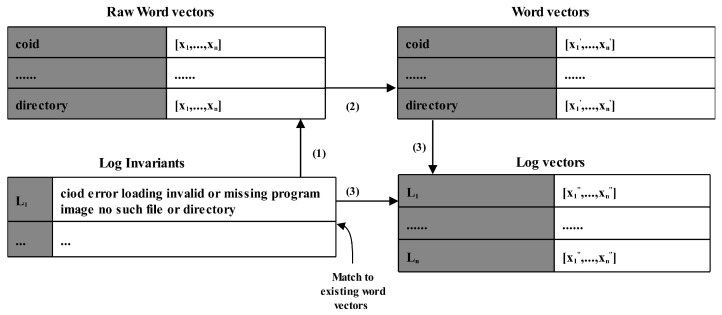
Example of log2vec.

**Figure 4 sensors-21-06125-f004:**
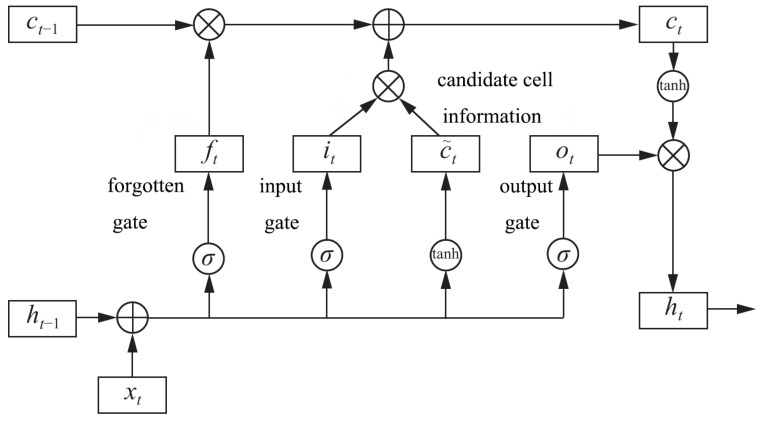
Lstm structure.

**Figure 5 sensors-21-06125-f005:**
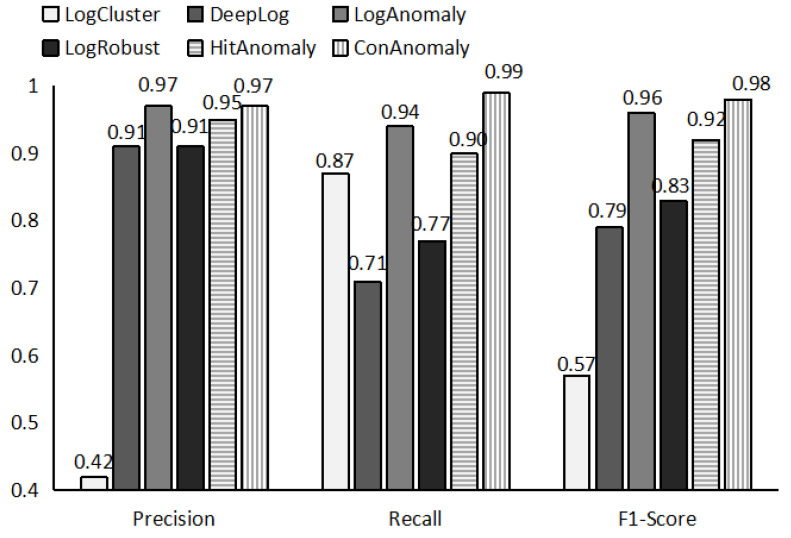
Evaluation on BGL dataset.

**Figure 6 sensors-21-06125-f006:**
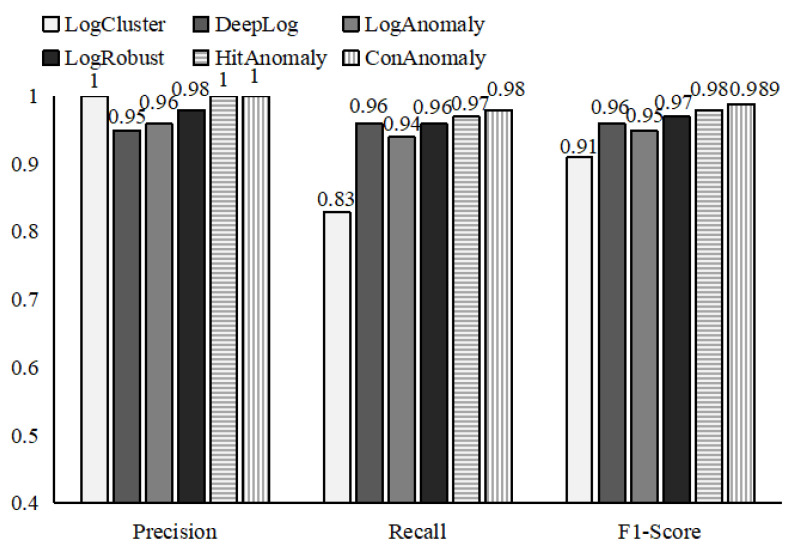
Evaluation on HDFS dataset.

**Figure 7 sensors-21-06125-f007:**
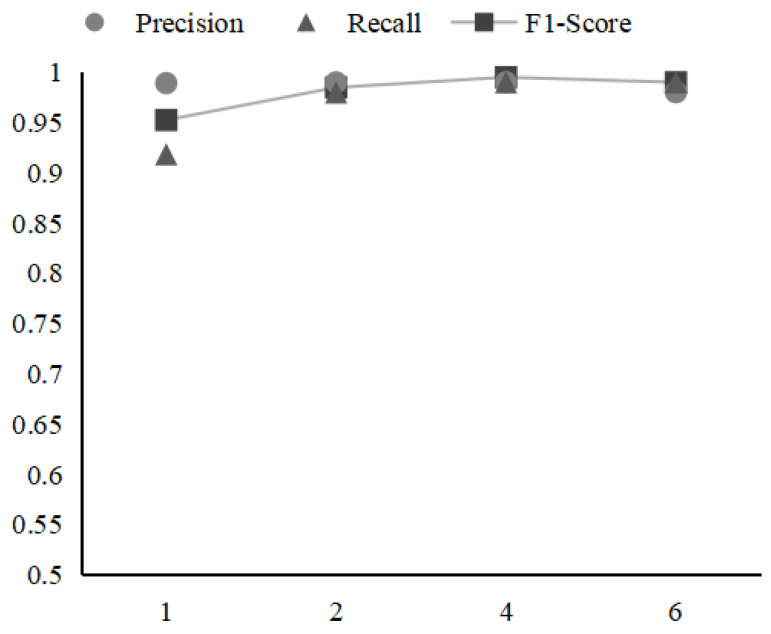
The impact of n-layer on the performance of ConAnomaly.

**Figure 8 sensors-21-06125-f008:**
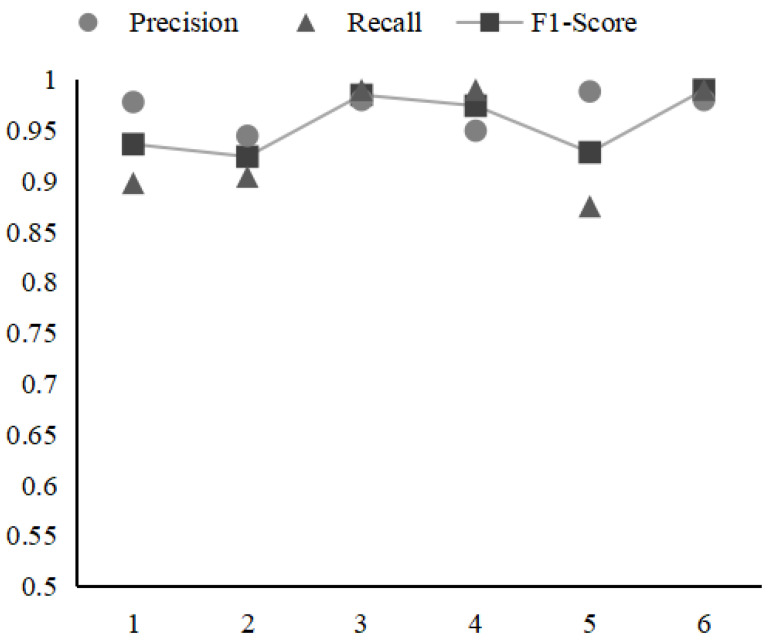
The impact of window size on performance of ConAnomaly.

**Figure 9 sensors-21-06125-f009:**
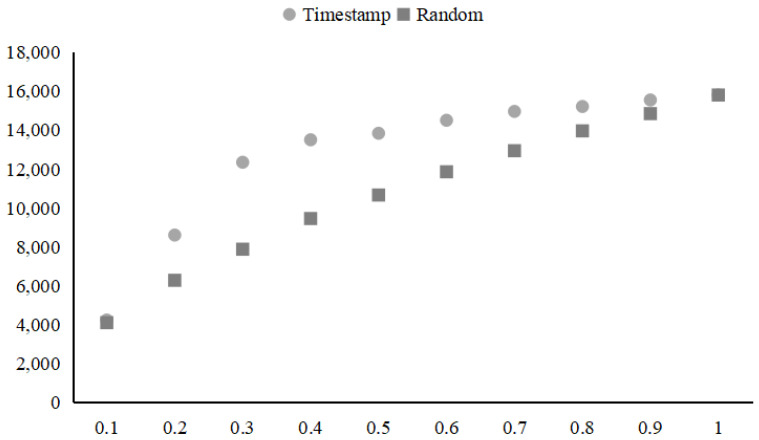
Log types distribution on HDFS dataset.

**Table 1 sensors-21-06125-t001:** The meaning of the filtered part of speech tag.

The Name of the Part of Speech Tag	Meaning
‘CC’	coordinating conjunction
‘TO’	‘to’
‘IN’	preposition/subordinating conjunction
‘MD’	modal (could, will)

**Table 3 sensors-21-06125-t003:** Partially digitized BGL logs.

Sequences Id	Log Sequences Based on the Fixed Window
s1	22 12 22 12 22 12 22 22 12 22 12 22 12 22 12 22 12 12 12 22
s2	168 168 168 168 168 168 168 168 168 168 168 168 168 168 168 168 168 168 168 168
s3	168 168 168 168 168 168 168 168 168 168 168 168 168 168 168 168 168 168 168 168
s4	168 168 168 168 168 168 168 168 168 168 168 168 168 168 168 168 168 168 168 168
s5	189 189 189 189 189 189 189 189 189 189 189 189 189 189 189 189 189 189 189 189
s6	201 149 201 149 201 149 201 149 201 149 201 149 201 149 201 149 201 149 201 149
s7	168 168 168 168 168 168 168 168 168 168 168 168 168 168 168 168 168 168 168 168
s8	168 168 168 168 168 168 168 168 168 168 168 168 168 168 168 168 168 168 168 168
s9	305 305 305 305 305 305 305 305 305 305 305 305 305 305 305 305 305 305 305 305

**Table 4 sensors-21-06125-t004:** Partially digitized BGL logs.

The Number in Table 3	The Log Template It Represents
12	(.*) microseconds spent in the rbs signal handler during (.*) calls. (.*) microseconds was the maximum time for a single instance of a correctable ddr.
22	(.*) total interrupts. (.*) critical input interrupts. (.*) microseconds total spent on critical input interrupts
149	external input interrupt (.*) (.*) (.*) tree receiver (.*) in resynch mode
168	gister: machine state register: machine state register: machine state register: machine state register: machine state register:
189	interrupt threshold...0
201	Lustre mount FAILED:(.*):point /p/gb1
305	program interrupt: unimplemented operation..0

**Table 5 sensors-21-06125-t005:** Log types distribution on HDFS dataset.

Percentage of Data	Divided by Timestamp	Divided in Random
0.1	4232	4097
0.2	8601	6275
0.3	12,344	7873
0.4	13,500	9451
0.5	13,835	10,660
0.6	14,503	11,855
0.7	14,959	12,938
0.8	15,212	13,959
0.9	15,547	14,846
1	15,802	15,802

**Table 6 sensors-21-06125-t006:** Statistics of unseen log types on HDFS dataset (the first row is training ratio).

	1%	10%	20%	50%
# **in training**	991	4201	6296	10,778
# **in testing**	15,719	14,878	13,896	10,549
# **unseen in training**	14,811	11,601	9506	5024
**F1-score**	0.95	0.97	0.98	0.98

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
