# Peer review of "ConAnomaly: Content-Based Anomaly Detection for System Logs"

_sensors, 2021, doi:10.3390/s21186125_

Round 1

Reviewer 1 Report

In the Reviewer’s opinion, this is a novelty work on the design of the anomaly detection method base on log sequence encoder and multi-layer long short-term memory network. Experimental results show the effectiveness of the proposed detector.

I have the following suggestions:

1) More classical anomaly detection method based on machine learning should be added into the introduction part to enrich the content.

2) Lots of new detectors can be considered in your future works, such as the matrix information geometry (MIG) detectors [1] and the subspace detectors [2]. please extending your conclusion by add these detectors:

[1] “MIG median detectors with manifold filter,” Signal Processing, vol. 188, p. 108176, 2021. [Online]. Available: https://www.sciencedirect.com/science/article/pii/S016516 8421002140.

[2] "Persymmetric Subspace Detectors With Multiple Observations in Homogeneous Environments," in IEEE Transactions on Aerospace and Electronic Systems, vol. 56, no. 4, pp. 3276-3284, 2020.

Author Response

We would like to thank you for your careful reading, helpful comments, and constructive suggestions. We have carefully considered all comments from the reviewer and revised our manuscript accordingly. Please see the attachment.

Reviewer 2 Report

Poor title!

plural of index is not indexes?

“got rid of”, this is a scientific work and you should not use these types of words

what package did you use to impelment LSTM, Keras or tenserFlow? In any condition, the code should be shared, and also If any of the above used you should mention that as your reference

Poor presentation Figure 1

ht-1, xt, t is not time? And should not be an index?

Where are the accuracy matrices? How did you evaluate the performance of the model?

In general, while going randomly over you manuscript, I see many grammatical and scientific errors.

Please make sure you proof-edit your manuscript extensively and make sure there is no error in the paper, especially mathematical formulation, especially pay attention to presentation. I will go over your paper in detail next time and I would reject it if there are errors.

By the way, I like the section of “Limitation of Previous Models”. You should follow similar structures for other parts. Also afterward you need to add study contributions.

Please highlight all changes and answer the points.

Author Response

We would like to thank you for your careful reading, helpful comments, and constructive suggestions. We have carefully considered all comments from the reviewers and revised our manuscript accordingly. Please see the attachment.

Round 2

Reviewer 1 Report

I don’t think this paper has fully been revised. The authors claim “Most machine learning models for anomaly detection cannot directly process text data”. It’s nonsense.

In my opinion, the presented method in this paper is an improvement of an existence algorithm. This method belongs to one of the machine learning methods. Most machine learning methods can be used to deal with the problem of anomaly detection of text data. Such as the classical PCA, SOM, SVM, K-means, and the deep learning methods, e.g., autoencoder, LSTM, etc. These methods can process the problem of anomaly detection in the specific case, respectively.

In addition, many detectors widely used in the field of signal processing, e.g., the matrix information geometry (MIG) detectors and the subspace detectors, these works, one of these methods is [1-2], have been confirmed that they are effective detection methods. The principle of these detectors can be referred.

[1] “MIG median detectors with manifold filter,” Signal Processing, vol. 188, p. 108176, 2021. [Online]. Available: https://www.sciencedirect.com/science/article/pii/S016516 8421002140.

[2] "Persymmetric Subspace Detectors With Multiple Observations in Homogeneous Environments," in IEEE Transactions on Aerospace and Electronic Systems, vol. 56, no. 4, pp. 3276-3284, 2020.

I think the authors must enrich the content of the introduction by adding these classical machine learning methods and lots of new detectors, MIG and subspace detectors in the field of signal processing. Otherwise, it is an incomplete work and is confusing.

Author Response

Thank you for your valuable suggestions, we have carefully considered all comments and revised our manuscript accordingly. Please see the attachment.

Reviewer 2 Report

addressed

Author Response

Thank you for your valuable suggestions.